# Climate Change Related Depression, Anxiety and Stress Symptoms Perceived by Medical Students

**DOI:** 10.3390/ijerph19159142

**Published:** 2022-07-27

**Authors:** Lukas Schwaab, Nadja Gebhardt, Hans-Christoph Friederich, Christoph Nikendei

**Affiliations:** Center for Psychosocial Medicine, Department for General Internal Medicine and Psychosomatics, Heidelberg University Hospital, 69115 Heidelberg, Germany; schwaab@stud.uni-heidelberg.de (L.S.); nadja.gebhardt@med.uni-heidelberg.de (N.G.); hans-christoph.friederich@med.uni-heidelberg.de (H.-C.F.)

**Keywords:** climate change, climate change-related mental burden, mental health, depression, anxiety, perceived distress, resilience factors, climate-related distress, climate anxiety

## Abstract

Climate change has drastic consequences on human physical and mental health. However, research on the psychological effects of climate change awareness is still inconclusive. To examine the mental burden posed by climate change awareness and potential resilience factors, *n* = 203 medical students were surveyed about their awareness of the implications of climate change. Furthermore, well-established mental health questionnaires (PHQ-9, GAD-7, PTSS-10, PSQ-20) were presented twice, in their original form and in a modified version to specifically ask about the respective psychological burden regarding climate change. For identification of potential resilience factors, measures for attachment style (RQ), structural abilities (OPD-SF), and sense of coherence (SOC-13) were used. The results of our study suggest that medical students in Germany have an increased risk to suffer from mental health problems and predominantly experience significant perceived stress in regard to climate change. However, the reported stress does not yet translate into depressive, anxious, or traumatic symptoms. Climate-related perceived stress correlates negatively with potential resilience factors preventing the development of mental disorders such as attachment style, structural abilities, and sense of coherence.

## 1. Introduction

The global climate crisis is a substantial threat to human existence and therefore must be regarded as a global emergency [1]. As assessed by international collaborations such as the *Lancet* Countdown on health and climate change [2], climate change has drastic consequences on human physical and mental health. Impairment of physical health associated with rising mean temperature is, for instance, reflected in heat-related deaths [3], adverse somatic effects of increased exposure to wildfires [4] and droughts [5], the spread of vector-borne diseases [6], and the growing number of humans affected by food insecurity and undernutrition [7]. Known effects on mental health are the psychological suffering arising from natural disasters associated with climate change [8] and the direct effects of rising temperature on mental well-being. The latter include changes in personality traits [9], intensified delinquency [10], and enhanced suicide rates [11]. Beyond this, the psychological effects of climate change awareness on mental well-being have sparked an intense discussion. The psychological burden and distress due to awareness of environmental change have thus far been conceptualized as solastalgia [12], environmental distress [13], climate grief [14], environmental melancholia [15], and climate anxiety. The most intensively discussed phenomena of mental health impairment related to climate change awareness at present are “eco-anxiety” and “climate anxiety”, or more specifically “climate change anxiety” [16], defined as “a chronic fear of environmental doom” [17]. 

While most authors agree that climate change awareness causes psychological distress, its exact impact on the development of mental health disorders, potential symptom load, and therefore clinical relevance are still a matter of debate. The aforementioned concepts have been comprehensively discussed in the literature and several instruments have been developed to measure the postulated effects, such as the Environmental Distress Scale (EDS) [13], a measure by Searle and Gow [18] depicting climate change distress and the Climate Anxiety Scale (CAS) [19]. However, divergent content validity, for example, the distinctiveness of the measured constructs from general measures of psychological distress, remains unclear. An attempt to adapt existing mental health measures that are widely used in the clinical setting to close this gap has not been made thus far. Additionally, a replication of the original factor structure of the CAS in a German sample in a recent study has not been successful [20]. 

People’s resilience has an influence on their reactions to potentially stressful events or circumstances such as climate change. There is no single factor shared by highly resilient individuals, but rather a complex and dynamic set of factors [21]. Resilience is influenced by biological, social, cultural, and psychological factors [22]. However, the present study focuses on resilience factors on an individual psychological level in comparison to the experienced psychological symptom burdens in question. In general, psychological resilience can be defined as the ability to maintain or regain mental health despite experiencing adversity [23]. Scientific research on psychological resilience factors when facing climate change is scarce. However, from trauma research, there do exist well-known resilience factors on an individual psychological level that are preventive for the development of trauma-related symptoms, namely a secure attachment style [24,25], structural personality functioning [26], and a pronounced sense of coherence [24,27].

The health care system’s role in relation to climate change is threefold: it is a relevant emitter that must reduce its greenhouse emissions [28,29], while at the same time it will be faced directly with the physical and mental consequences of climate change. Furthermore, its employees represent a well-trusted part of society [30] and are skilled communicators who could act as mediators between science, policy, and practice [28]. Accordingly, climate change is highly likely to shape the medical profession in the future. The ClimAttitude Study [31] investigated German medical students’ attitudes towards climate change. Participants were in their final year of studies and represent a new generation of physicians. While final-year medical students showed a high awareness of the consequences climate change has on human health and accepted their individual responsibility as a part of society, they were unsure about the medical profession’s duty to act as an ecological role model. While medical students have shown an elevated risk of mental health problems [32,33], the future medical doctors’ own psychological burden related to climate awareness remains unclear thus far.

Considering the important role the health care system plays in relation to climate change and the high levels of awareness of the consequences of climate change that medical students in their final year displayed in the ClimAttitude study [31], the present study aims to analyze data from German medical students in the clinical part of their studies. The study focuses on three goals: first, confirming medical students’ high levels of awareness of the global climate crisis and its implications. Second, applying widely used, validated mental health questionnaires to examine the students’ mental health as well as adapting these instruments to evaluate the impact of climate change on their mental health, and third, detecting potential resilience factors by taking personality factors into account.

## 2. Materials and Methods

### 2.1. Study Design, Participants and Procedure

The presented cross-sectional questionnaire-based study is part of the CliMental project which aims to investigate the effect of climate change on mental health in different subgroups of German society. For the purpose of the current study, medical students in the third, fourth, and fifth years of their clinical studies were recruited. All students (*n* = 249) taking their final exams in Psychiatry and Psychosomatic Medicine at the University of Heidelberg between May and December of 2021 were asked to participate. Students were informed orally by the first author about the purpose of the study after the exam and stayed in the room to fill out the paper-based questionnaire if they agreed to participate. Participants gave written informed consent. Sociodemographic characteristics, attitude towards climate change, different aspects of mental health, and potential resilience factors were included in the questionnaire. The participants needed approximately 15 to 20 min to complete the questionnaires. No compensation was paid for participation. The study was approved by the ethics committee of the Medical Faculty of the University of Heidelberg (S-450/2021) and was in line with the Declaration of Helsinki.

### 2.2. Measurement of Climate Change Awareness

In the eighth round of the European Social Survey (ESS), conducted in 2016, people across Europe were surveyed on social issues, including their thoughts and worries about climate change [34]. The ten questions of the ESS accounting for attitudes towards climate change were used in this study to map the medical students’ awareness of climate change and its implications. First, participants were asked on a four-point Likert scale, ranging from “definitely” to “certainly not”, if they believed climate change existed. In addition, they were questioned on a five-point Likert scale, ranging from “very much” to “never”, about how much they had been thinking about climate change until this point. Moreover, they were asked on a six-point Likert scale, ranging from “extremely worried” to “doesn’t exist”, how worried they were about climate change. In a further step, they were asked on a ten-point Likert scale (0 = not at all; 10 = very much), how much they felt personally responsible to try to reduce climate change. On a ten-point Likert scale (0 = very unlikely; 10 = extremely likely) students were asked how likely the participants thought it was, that their personal effort or an effort made by society would help to reduce climate change, and how likely it was, that enough countries will put appropriate measures in place to reduce climate change. Lastly, they were asked (“yes” or “no”), if they were currently more worried about climate change in comparison with one, three, and five years ago.

### 2.3. Psychometric Measures for Mental Health

Well-established mental health questionnaires were used for measures of depression (PHQ-9), anxiety (GAD-7), traumatic stress (PTSS-10), and perceived stress (PSQ-20). Each item of these measures was presented twice in succession before continuing with the next item. First, the items were presented in the original version of the respective questionnaire. Second, the items were modified to specifically ask about the respective psychological burden regarding climate change. In the following analysis, these climate change-specific questionnaires will be referred to as PHQ-9-C, GAD-7-C, PTSS-10-C, and PSQ-20-C.

***PHQ-9****(**Brief Patient Health Questionnaire**)**.*** For diagnostics of depressive disorders, this study used the PHQ-9 module of the PHQ-D [35], which is a reliable and valid measure to make a criteria-based diagnosis and assess the severity of depressive symptoms [36]. 

***GAD-7****(**Generalized Anxiety Disorder Scale-7**)**.*** The second PHQ-D module used in this study is the GAD-7, which is a valid and efficient tool for screening for general anxiety disorder [37]. 

***PTSS-10 (Posttraumatic Stress Scale-10).*** The PTSS-10 is a German short scale [38] used for screening for post-traumatic stress disorder according to DSM-IV that is derived from the original English scale by Breslau et al. [39].

***PSQ*** (***Perceived Stress Questionnaire***)**.** The *PSQ* is used to assess the subjective perception, evaluation, and further processing of stressors [40]. The validated German version used here is shortened to 20 items [41] comprising four subscales of five items each: worries, tension, demands, and joy (to be inverted). Cut-off scores for low, moderate, or high levels of perceived stress were derived from the representative study of the German population conducted by Kocalevent et al. [42].

### 2.4. Measures for Potential Resilience Factors

For the identification of potential resilience factors, three questionnaires were used as measures for attachment style (RQ), structural abilities (OPD-SF), and sense of coherence (SOC-13). 

***RQ****(**Relationship Questionnaire**)**.*** The RQ is a four-item measure, designed to assess attachment style [43]. This study used the German translation and modification according to Asendorpf et al. [44].

***OPD-SF****(**OPD Structure Questionnaire**)**.*** The short version of the OPD Structure Questionnaire (OPD-SFK) was developed by Ehrenthal et al. [45] to assess structural abilities according to OPD-2, previously only accessible through OPD (Operationalised Psychodynamic Diagnostics) interviews.

SOC (Sense of Coherence Scale). 

The SOC scale captures the concept of a sense of coherence, which can be described as a helpful personality trait in leading a healthy and content life in the face of external stressors. It consists of three dimensions, namely comprehensibility, manageability, and meaning [46]. The German short version SOC-13 used here has a high internal consistency and correlates strongly with the long version of the scale [47]. 

### 2.5. Statistical Analysis

Data were analyzed using the software package IBM SPSS Statistics 28 (SPSS Inc., Chicago, IL, USA). In order to describe the characteristics of the study group (demographic data, such as age, sex, etc.) as well as their attitude towards climate change, descriptive statistics were generated. Furthermore, to measure the internal consistency of the adapted questionnaires (PHQ-9-C, GAD-7-C, PTSS-10-C, PSQ-20-C), Cronbach’s alpha was calculated. To evaluate students’ mental health and the impact of climate change on their mental health, we calculated the individual values that participants scored on the original and adapted questionnaires by applying the validated cut-offs for different levels of symptom severity and assessing the frequency of these degrees of psychological burden among the study group. A *p*-value < 0.05 (two-tailed) was considered statistically significant. The datasets analyzed during the current study are available from the corresponding author on reasonable request.

## 3. Results

### 3.1. Sample

Table 1 depicts the characteristics of the participating students. A total of *n* = 249 students were invited to take part in the study. Of these students, *n* = 216 (response rate: 87%) participated. Thirteen of the participating students could not finish filling out the questionnaire due to time limitations (having to leave the room for the next group taking the exam) and were therefore excluded. Consequently, results were calculated for *n* = 203 (82%) students’ data. Three (1%) of the items measuring attitude towards climate change were missing values. Those were not replaced, as the questionnaires’ results are only interpreted in a descriptive manner. 

For the psychometric measures and the personality factors, 264 (1%) values were missing. Those were replaced by the mean of the item calculated from the available data points, as this method does not overestimate the data’s variance and results are comparable to those obtained when employing multiple imputation methods [48]. 

### 3.2. Attitude towards Climate Change

Students perceive climate change as real, as 88% stated that they believe the climate is changing, 11% stated it is probably changing and only one student (1%) stated that the climate is probably not changing. Additionally, students in the sample showed awareness of the topic, as 60% of the participants declared that they think about it very often or often, and the same number of students stated they are extremely worried or very worried about it. These worries about climate change seemingly increased within the last years: 85% of the sample indicated worrying more about climate change today than five years ago, 75% worried today more than three years ago, and 33% worried today more than last year. On a scale ranging from 0 (“not at all”) to 10 (“absolutely”), students judged a societal effort to restrict energy consumption as the likelier scenario to mitigate climate change (*M* = 6.04, *SD* = 2.44) in comparison to an individual effort to reduce one’s own energy consumption (*M* = 3.39, *SD* = 2.50). On average, students did not think it to be likely that a sufficient number of countries would put measures into place that are in time to effectually mitigate climate change (*M* = 2.67, *SD* = 2.30).

### 3.3. Students’ Mental Health and Climate Change’s Role in It

Students’ general mental health and symptom burden related to climate change were assessed for depression (PHQ-9 and PHQ-9-C), anxiety (GAD-7 and GAD-7-C), traumatic symptoms (PTSS-10 and PTSS-10-C), and perceived stress (PSQ-20 and PSQ-20-C). Raw scores were transformed into clinically meaningful categories relying on the standard samples reported in the respective manuals or publications. For the PSQ-20 and the PSQ-20-C, cut-off scores were derived from a representative sample of the German population reported in a study by Kocalevent et al. [42]. The descriptive results are shown in Table 2. As can be seen, PHQ-9, GAD-7, and PTSS-10 were sensitive to general symptom burden, whereas the respective climate change-related versions did not measure any clinically significant symptom burden. Both versions of the questionnaires showed good to excellent internal consistencies [Cronbach’s alpha; (PHQ-9) = 0.85, (PHQ-9-C) = 0.88, (GAD-7) = 0.85, (GAD-7-C) = 0.83, (PTSS-10) = 0.87, (PTSS-10-C) = 0.90]. In contrast, 45 participants (23%) reported moderately or highly alleviated levels of stress regarding climate change in the PSQ-20-C in comparison to 55 (27%) participants who reported moderately or highly alleviated levels of general stress in the PSQ-20. Internal consistency was calculated to be acceptable for the general version and excellent for the climate change-related version [(PHQ-20) = 0.76, (PHQ-20-C) = 0.90].

Differences between reported scores in the original and modified climate change-related versions of the questionnaires were evaluated for statistical significance using the raw scores derived from the sample. The Kolmogorov-Smirnov test was used to test for the normal distribution of the data. Data was not normally distributed for any of the questionnaires [PHQ: D(203) = 0.15, *p* < 0.001; GAD: D(203) = 0.12, *p* < 0.001; PTSS: D(203) = 0.13, *p* < 0.001; PSQ: D(203) = 0.09, *p* = 0.001], hence the Wilcoxon-Signed-Rank test being employed to ensure the robustness of results. The score of the original version was significantly higher than for the modified climate change specific version for PHQ [T(203) = −11.56, *p* < 0.001], GAD [T(203) = −10.83, *p* < 0.001] and PTSS [T(203) = −11.71, *p* < 0.001], with a large effect size calculated using Cohen’s d [d(PHQ) = 1.14, d(GAD) = 1.09, d(PTSS) = 1.14]. For the PSQ, however, scores did not differ significantly [D(203) = −0.11, *p* = 0.912] with an effect size of d(PSQ) = −0.04, indicating that the overall stress level and the climate change-specific stress level were equally high.

### 3.4. Exploratory Factor Analyses

As we found indications for high levels of climate-related stress in the PSQ-20-C, we further examined the modified climate change-related version of the PSQ, running a factorial analysis to explore its structure. Items were well suited for factor analysis in both versions of the questionnaire [KMO(PSQ-20) = 0.95, KMO(PSQ-20-C) = 0.91], and significantly differed from its identity matrix as tested for with the Bartlett test of sphericity (PSQ-20: 2 = 5388.03, df = 190, *p* < 0.001; PSQ-20-C: 2 = 2128.71, df = 190, *p* < 0.001). The original publication [41] reports four subscales with five items each for the PSQ-20: worries, tension, demands, and joy. Conducting a confirmatory analysis, the hypothesized model showed an acceptable fit for the PSQ-20 with the respective fit indices being Comparative Fit Index (CFI) = 0.93, Tucker-Lewis Index (TLI) =  0.91, Akaike Information Criterion (AIC) = 19,612.26, Root Mean Square Error of Approximation (RMSEA) = 0.072, 90% CI [0.066, 0.079], and Standardized Root Mean Square Residual (SRMR) = 0.061. For the PSQ-20-C, however, the model did not show a satisfactory fit with a Comparative Fit Index (CFI) = 0.83, Tucker-Lewis Index (TLI) = 0.80, Akaike Information Criterion (AIC) = 20,736.95, Root Mean Square Error of Approximation (RMSEA) = 0.107, 90% CI [0.101, 0.113], and Standardized Root Mean Square Residual (SRMR) = 0.121.

Consequently, an exploratory factor analysis with oblique rotation was run for the PSQ-20-C to further explore its structure (Appendix A). Parallel analysis [49] recommended two factors. Inspection of the scree plot showed that there were only two factors with an Eigenvalue > 1, so the analysis was run for two factors. Inspection of the loading matrix revealed that the first factor comprised all items of the original subfactors worries, tension, and demands except for items 1 (“I feel rested”) and 6 (“I feel calm”), which originally belonged to the subscale tension, and item 19 (“I have enough time for myself”), which originally belonged to the subscale demands. These and the subscale joy, which has to be inverted as a whole formed the second factor. Therefore, all the inverted items belonged to one factor. The factors correlated with r = 0.253 (*p* < 0.001) for the inverted version of the second factor, which indicated that a higher rating for the items of the original subscales worries, tension, and demands correlates with lower ratings of joy and of items 1, 6 and 19. The first factor showed a lower mean (*M* = 19.93, *SD* = 6.96) than the second factor in its inverted form (*M* = 23.02, *SD* = 5.65), which implied that thinking about climate change reduces positive feelings to a higher degree than it evokes negative feelings. 

### 3.5. Correlation of Climate Change Related Stress with Personality Factors

Table 3 depicts participants’ scores on the PSQ-20-C. As can be seen, there was no significant correlation with the RQ-Other dimension but a significant correlation with lower scores on the RQ-Self dimension. Furthermore, PSQ-20-C was significantly associated with higher scores on the OPD-SF, and with lower scores on the SOC-13. 

## 4. Discussion

To the best of our knowledge, the presented study is the first to explore the mental health burden of climate change on medical students by using adapted versions of widely used and validated mental health questionnaires. Participants showed a high awareness of climate change and substantial worries about its implications. Assessing medical students with validated questionnaires, 42% showed symptoms of depression and 50% symptoms of general anxiety. 10% had high enough scores on the PTSS-10 scale to suspect a clinically significant traumatic symptom burden. 27 % of students reported moderate or high levels of perceived stress. When assessing climate-related depression, anxiety, and traumatic stress, few of the participants showed signs of depression, general anxiety disorder, or traumatic stress associated with climate change. However, 23% of participants showed moderate or high levels of perceived stress as a consequence of climate change. This was more prominent in participants with reduced structural abilities and a lower sense of coherence. 

Regarding participants’ characteristics, 58% of participants in this study were extremely or very worried about climate change, in comparison to 43% of the German general population as reported by the eighth round of the European Social Survey [24]. This confirms the hypothesis made by Bugaj et al. in the ClimAttitude study [31] that medical students show high awareness of the threat that climate change poses to our planet and especially humanity. Two factors that make medical students more susceptible to worries about climate change compared to the general population may be their young age as well as their relatively high level of education, as younger generations and people with post-secondary education have been shown to be more aware and concerned about climate change [50,51]. Our results also show that the concern about climate change has been increasing among participants over the last five years. 33% of the participants stated that they were more worried about climate change than one year ago, 75% more than three years ago, and 90% more than five years ago. This could be, among other influences, a result of the increased media coverage of climate change-associated news in recent years [1].

60% of the participants had scores of 7 or higher on an 11-point Likert scale from 0 (not at all) to 10 (a great deal) asking how much they feel personally responsible to reduce climate change. However, when asked if the participants believed reducing their personal footprint would help reduce climate change, almost half the participants had scores of 3 or less on an 11-point Likert scale from 0 (not at all likely) to 10 (extremely likely). This can be interpreted as a sign of low self-efficacy, suggesting that medical students have not yet fully realized the substantial amount of emissions they will be responsible for as a part of the healthcare sector, as well as their potential for decreasing these emissions. Examples of possible actions they can take in the future include counseling their patients on environmentally friendly behavior, promoting health, avoiding medical overuse, and giving preference to treatment options and medical technologies with a lower environmental impact [52]. As future medical professionals, the participants will hold a central role in decreasing healthcare’s sizeable contribution to global greenhouse gas emissions, a need that urgently has to be addressed by all of the healthcare systems around the globe [53]. 

Regarding depression, anxiety, and general perceived stress scores, medical students showed a higher percentage of clinically significant symptoms in comparison to prevalence rates reported by population-based surveys [42,54,55]. This is in line with previous reports on mental health among medical students indicating an alleviated risk for mental health problems in this population [32,33]. Traumatic stress symptoms, however, were not increased [38]. 

The general symptom burden and participants’ worries about climate change expressed in the first part of the questionnaire did not reflect in high scores on the PHQ-9-C, GAD-7-C, and PTSS-10-C. Only 3% of participants showed at least mild symptoms of depression associated with climate change, compared to 42% with symptoms of depressive disorder on the PHQ-9. 10% showed at least low levels of anxiety associated with climate change, compared to 50% with increased levels of anxiety on the GAD-7. None of the participants showed moderate or severe symptoms of depressive disorder or general anxiety disorder in relation to climate change. On the PTSS-10-C scale, none of the medical students scored high enough to suspect a traumatic symptom burden in relation to climate change. Hence, the results of our study suggest that the stated worries about climate change do not result in a clinical impairment on a psychometric level within the study group on these instruments. This is in line with several studies suggesting that comparatively high awareness of climate change, which could be read as “climate anxiety”, may not have a clinically significant negative effect on mental health and could instead lead to adaptive reactions such as pro-environmentalist behavior [20,56]. With regard to the experience of trauma symptoms in relation to climate change, it should be noted that the participants were not witnesses of natural disasters due to their geographical location and therefore do not fulfill the strict etiological requirements for trauma. However, it is known from obstetrics, for example, that anticipatable traumatic events such as the birth process may be linked to so-called pre-traumatic stress conditions [57]. This may raise the question of whether climate change and the associated related changes in our habitat could also be understood as a pre-traumatic stress condition. The phenomenon of secondary traumatization has proven to lead to the development of traumatic stress symptoms due to another person’s mere report of a traumatic event through empathizing and visualizing the trauma without directly experiencing the respective traumatic event [58]. In line with these findings, the empathizing and visualizing of future climate-related events could result in a pre-traumatic stress condition as well. However, the results of this study indicate that the effects of climate change, especially from our perspective in the northern hemisphere, do not yet seem threatening enough for this kind of pre-traumatic stress.

In contrast, participants’ perceived stress associated with climate change documented via PSQ-20-C was equally as high as their perceived stress on the PSQ-20: 18% of the sample reported a moderate level of perceived stress, 5% a high level of perceived stress. As the subsequent factorial analysis of the PSQ-20-C suggests, it does not seem to measure the same construct as the PSQ-20 does. The two emerging factors in the explanatory factorial analysis seem to allow a distinction between negative feelings and thoughts in relation to climate change (first factor), and the lack of positive feelings and thoughts (second factor). Of these two, the lack of positive feelings and thoughts in relation to climate change seems more pronounced. This may contribute to an explanation why the perceived stress due to climate change awareness reported in this study did not lead to a higher rate of symptoms of depression, anxiety, or trauma: the perceived stress rather translates into an absence of positive feelings than into negative feelings to a clinically significant extent. Interestingly, none of the instruments existing so far to measure climate change distress or anxiety [14,19,20] take positive emotions into account. Further research might be able to determine if the absence of positive feelings represents an important characteristic in understanding the construct of climate change awareness. 

In sum, participants in this study showed a high prevalence of symptoms of depression, anxiety, and perceived stress. When asked to consider the symptomatic burden with respect to climate change, they reported considerable levels of perceived stress, but not of depression, anxiety, or traumatic symptoms. Alleviated levels of stress are a well-known risk factor for the development of mental disorders [41]. As worries about climate change have steadily increased over recent years in the present sample, and its participants are relatively young, one may expect the continuously rising level of perceived stress to be detrimental to mental health and eventually lead to disorders. The assumption, that high awareness of the implications of climate change does not have a clinically significant negative effect on mental health, may therefore not be true in the future if it is accompanied by a high level of perceived stress due to climate change. The correlation between perceived stress in the PSQ-20-C and the medical students’ sense of coherence suggests that the issue is still subject to a sense of controllability for the participants. As the effects of the climate catastrophe progress, it is to be expected that this feeling of controllability will break down and symptoms of hopelessness and helplessness, feelings that have long been associated with the development of depression and anxiety disorder [59], will be experienced.

Participants’ scores on the PSQ-20-C did not correlate significantly with a robust attachment style towards others (*RQ others*), but negatively with a robust attachment style with oneself (*RQ self*), positively with the OPD-SF (whose items are inversed), and negatively with the SOC-13. Therefore, participants with a higher level of perceived stress due to climate change tend to show a less secure attachment style, less structural abilities, and a less pronounced sense of coherence, suggesting that those personality factors could represent resilience factors against heightened stress responses in relation to climate change. These correlations are in line with research on the development and etiology of mental disorders, to whose development a heightened stress response to climate change awareness may contribute. A robust attachment style is known to be correlated with a decreased risk for the development of depressive disorders and post-traumatic symptoms [60,61,62,63]. Likewise, there is a negative correlation between a sense of coherence and post-traumatic, as well as depressive symptoms [64,65,66]. For structural abilities as measured by the OPD-SF, a higher impairment was shown for clinical, in comparison to population-based samples [67,68]. These findings are in line with previous research on psychological resilience factors for secondary traumatization [24,25,26,27], implying the possibility for a certain comparability of the mechanisms underlying the stress responses in both cases.

### Strengths and Limitations

The presented study is explorative in character, as it employs questionnaires that are validated and widely used in clinical settings to measure depression, anxiety, traumatic symptom burden, and perceived stress in an adapted version to measure the extent of the respective symptoms when thinking about climate change. The response rate among students was high. All data was generated from students attending the same program at the same university and therefore should be handled with care if translated into another context, as the prevalence of mental health disorders and attitudes towards climate change may vary between regions or countries. Due to the cross-sectional nature of this study, it is important to be aware of its predictive limitations: In order to establish a true cause and effect relationship between climate change and mental health, future longitudinal data is needed. The current study can lay a basis for such longitudinal observations. As with many psychological studies, methodological limitations also include the possibility of bias which may have resulted from the ordering of questions and from the use of uni-directional scales. Additionally, the context for the completion of the questionnaires might have had an effect on the results of this study, as previous research has shown that exam situations have a significant effect on mood parameters and salivatory cortisol levels of medical students [69]. Fatigue following the completion of their exam could also have an influence on the participants’ results. Another potential distortion of the results of this study is the SARS-CoV-2 pandemic taking place during the survey period, as it has a negative effect on mental health and eclipsed other societal issues such as the ongoing climate crisis at the time of the survey. It is important to note that the climate change-related questionnaires used in this study were adapted to investigate the psychological burden of climate change, although they have not been validated for this inquiry. However, the measures showed good properties, and previous research, for example by Grosse-Holz et al. [70], could be used as an example of how to successfully adapt mental health questionnaires. 

## 5. Conclusions

The results of our study suggest that medical students in Germany have an increased risk of mental health problems and experience significant stress as measured with the PSQ-20-C in relation to climate change. However, the reported stress does not translate into depressive, anxious, or traumatic symptoms in relation to climate change as measured by clinical questionnaires. Climate-related perceived stress correlated negatively with potential resilience factors that play a role in the prevention of the development of mental disorders such as attachment style, structural abilities, and sense of coherence. It has yet to be seen if the high levels of climate change perceived stress depicted in the PSQ-20-C correlate with high levels of emotional and functional impairment or if the perceived stress might lead to positive coping mechanisms such as pro-environmentalism. To avoid feelings of hopelessness, especially in young adults who show a substantial worry about climate change, immediate climate action by global governments as well as offers of assistance on how to cope with the threatening situation and the strengthening of resilience is required. Additionally, to further research the nature and course of the psychological burden of climate change, longitudinal prospective and qualitative studies are needed.

## Figures and Tables

**Table 1 ijerph-19-09142-t001:** Sample Characteristics: Demographic data from *n* = 203 surveyed medical students.

*Item*	*Specification*	*n*	%
Sex	Male	85	42%
	Female	117	58%
Age	*M/SD*	25.21	3.71
Nationality	German	179	88%
	Non-German European	10	5%
	Non-European	12	6%
	Missing	2	1%
Currently in psychotherapy	Yes	12	6%
	No	191	94%
Previously in psychotherapy	Yes, once	26	13%
	Yes, multiple times	8	4%
	No	169	83%
Currently taking psychotropic medication	Yes	8	4%
	No	195	96%
Diagnosed with mental disorder	Yes	19	9%
	No	183	90%
	Missing	1	1%

**Table 2 ijerph-19-09142-t002:** Students’ general mental health and symptom burden related to climate change (*n* = 203).

Mental Health Disorder (Measure)	Original Version		Modified Climate Change Related Version	
	*n*	%	*n*	%
Depressive Disorder (PHQ-9; PHQ-9-C) ^a^				
No symptom burden	117	58%	198	97%
Mild symptoms	60	30%	3	1%
Moderate symptoms	15	7%	1	1%
Severe symptoms	11	5%	1	1%
General Anxiety Disorder (GAD-7; GAD-7-C) ^b^				
No symptom burden	103	50%	184	90%
Low levels of anxiety	71	35%	18	9%
Moderate levels of anxiety	24	12%	1	1%
High levels of anxiety	5	3%	0	0%
Traumatic symptom burden (PTSS-10; PTSS-10-C) ^c^				
No significant traumatic symptom burden	183	90%	202	99%
Positively screened for traumatic symptom burden	20	10%	1	1%
Perceived Stress (PSQ-20; PSQ-20-C) ^d^				
Average level of perceived stress	148	73%	158	77%
Moderate level of perceived stress	41	20%	36	18%
High level of perceived stress	14	7%	9	5%

*Annotation.*^a^ Brief Patient Health Questionnaire (PHQ-9, PHQ-9-C) cut-offs: 5–9 = mild symptoms; 10–14 = moderate symptoms; ≥15 = severe symptoms. ^b^ Generalized Anxiety Disorder Scale (GAD-7, GAD-7-C) cut-offs: 5–9 = low levels of anxiety; 10–14 = moderate levels of anxiety; ≥15 = high levels of anxiety. ^c^ Posttraumatic stress scale (PTSS-10, PTSS-10-C) cut-off: ≥35 = suspected traumatic symptom burden. ^d^ Perceived Stress Questionnaire (PSQ-20, PSQ-20-C) cut-offs: >*M* + 1 *SD* = moderate level of perceived stress; >*M* + 2 *SD* = high level of perceived stress (values for *M* and *SD* from Kocalevent et al. [42].

**Table 3 ijerph-19-09142-t003:** Correlations of the PSQ-20-C and the two factors found in the explanatory factor analysis of the questionnaire with the potential resilience factors attachment style (RQ), structural abilities (OPD-SF), and sense of coherence (SOC-13). Correlations were calculated based on the raw (inverted) scores of the sample.

	Attachment Style (RQ Self)		Attachment Style (RQ Other)		Structural Abilities (OPD-SF) ^a^		Sense of Coherence (SOC-13) ^b^	
	*r*	*p*	*r*	*p*	*r*	*p*	*r*	*p*
PSQ-20-C	−0.159 *	0.023	−0.118	0.094	0.389 **	<0.001	−0.375 **	<0.001
PSQ-20-C *factor 1*	−0.083	0.238	−0.025	0.722	0.343 **	<0.001	−0.305 **	<0.001
PSQ-20-C *factor 2*	−0.179 **	0.006	−0.178 *	0.007	0.267 **	<0.001	−0.289 **	<0.001

*Annotation.*^a^ A positive correlation indicates that the higher the level of perceived stress, the lower the structural abilities. ^b^ a negative correlation indicates that the higher the perceived stress, the lower the sense of coherence. * *p* < 0.05, ** *p* < 0.01.

## Data Availability

Data will be released upon reasonable request to researchers after contacting the corresponding author.

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
