# Peer review of "Climate Change Related Depression, Anxiety and Stress Symptoms Perceived by Medical Students"

_ijerph, 2022, doi:10.3390/ijerph19159142_

Round 1

Reviewer 1 Report

The study presented in the article is of great interest. It provides insights into how medical students in a Western context (Germany) experience climate change and makes a case about the importance of addressing this experience from a psychological perspective. However, the theoretical background is inadequate, especially in respect to the concepts of resilience and adaptability, there is no clear justification of the methodological design/selection of so many questionnaires, and there are conceptual leaps regarding the connection between the students' individual mental state and their psychological relation to climate change. The study could have been more targeted, theoretically grounded and connected to medical students' self perception/psychological state, future profession, locality and privilege (not directly affected by climate change). Please also see my comments included in the text. Thorough proofreading is required. Overall, a promising approach that needed further development and more attention to detail.  

Author Response

Thank you for your evaluation as well as your valuable comments included in the text. We proofread the manuscript again and conducted language edits and improved the scientific style as requested. To exhibit these changes, we have attached the revised manuscript. 

The local ethics committee was specified:

“The study was approved by the ethics committee of the Medical Faculty of the University of Heidelberg (S-450/2021) and was in line with the Declaration of Helsinki.“ (lines 108-110)

Point 1:"The theoretical background is inadequate, especially in respect to the concepts of resilience and adaptability"

We added a more thorough explanation of the concept of resilience employed in our study, defining and explaining why we are talking about resilience factors on an individual psychological level while acknowledging that resilience factors on a biological, social and cultural level are of importance as well:

“People’s resilience has an influence on their reactions to potentially stressful events or circumstances such as climate change. It is not a single factor shared by highly resilient individuals, but rather a complex and dynamic set composed of varying factors [22]. Resilience is influenced by biological, social, cultural and psychological factors [23]. However, the present study focuses on resilience factors on an individual psychological level in comparison to the experienced psychological symptom burdens in question. In general, psychological resilience can be defined as the ability to maintain or regain mental health despite experiencing adversity [24].” (lines 59-66)

Point 2: "there is no clear justification of the methodological design/selection of so many questionnaires, and there are conceptual leaps regarding the connection between the students' individual mental state and their psychological relation to climate change."

Additionally, we explained our rationale in choosing the questionnaires employed in this study:

“Scientific research on psychological resilience factors when facing climate change is scarce. However, from trauma research, there do exist well-known resilience factors on an individual psychological level that are preventive for the development of trauma related symptoms, namely a secure attachment style [25, 26], structural personality functioning [27] and a pronounced sense of coherence [25, 28].” (lines 66-70)

Concerning sense of coherence, we modified the given explanation to point out the link between this respective concept and mental health:

“The SOC scale captures the concept of a sense of coherence, which can be described as a helpful personality trait in leading a healthy and content life in the face of external stressors. It consists of three dimensions, namely comprehensibility, manageability and meaning [47]. (lines 166-169)

In the discussion, we provide an additional potential framework for examining resilience factors:

“These findings are in line with previous research on psychological resilience factors for secondary traumatization [25-28], implying the possibility for a certain comparability of the mechanisms underlying the stress responses in both cases.” (lines 441-443)

To clarify why we chose to evaluate a potential traumatic symptom load associated with climate change in a study group that most likely hasn’t experienced any catastrophic climate events due to their geographical background, we added a paragraph about pre-traumatic stress conditions and potential connections to climate change to the “Discussion” section:

“However, it is known from obstetrics, for example, that anticipatable traumatic events such as the birth process may be linked to so-called pre-traumatic stress conditions [57]. This may raise the question of whether climate change and the associated related changes in our habitat could also be understood as a pre-traumatic stress condition. The phenomenon of secondary traumatization has proven to lead to the development of traumatic stress symptoms due to another person’s mere report of a traumatic event through empathizing and visualizing the trauma without directly experiencing the respective traumatic event [70]. In line with these findings, the empathizing and visualizing of future climate related events could result in a pre-traumatic stress condition as well. However, the results of this study indicate that the effects of climate change - especially from our perspective in the northern hemisphere - do not yet seem threatening enough for this kind of pre-traumatic stress.“ (lines 365-376)

Reviewer 2 Report

How long does it take to complete the assessment tools you administered? That could be helpful information in the 2.1 session to clarify your procedure more. 

Section 2.3. I am a little unsure of the connection between the ESS study in 2016 and the assessments you use for this specific study? Please clarify more. 

Line 127 - did you mean to say item or items?

Did you administer these assessments back to back at one setting? Line 126-128. My concern is issues with test-retest reliability and fatigue of exams, then assessments of the subjects. That could be a significant limitation of this study. 

Line 184 - please further describe and clarify the reason you replaced missing values by the mean?

Lines 273-281 may be better placed in the discussion section instead of the results. You are using result data to then offer other ideas and future research. 

Line 319 - you state "over the last years." I would put in the number of years. One item that you may need to also discuss is that the COVID pandemic may have played a role in some of this change in views of climate change and overall distress in society. That may also be a limitation in the data. 

Line 417. The assessments you used in this study have been validated. But did you thoroughly validate y our modified versions of those assessments? 

Author Response

Thank you for your evaluation and valuable comments. We proofread the manuscript again and conducted language edits and improved the scientific style as requested. To highlight these changes, we have attached the revised manuscript. 

Point 1: How long does it take to complete the assessment tools you administered? That could be helpful information in the 2.1 session to clarify your procedure more. 

Response 1: Thank you for the remark. We added the time it approximately took to complete our survey:

“The participants needed approximately 15 to 20 minutes to complete the questionnaires.” (lines 106-107)

Point 2: Section 2.3. I am a little unsure of the connection between the ESS study in 2016 and the assessments you use for this specific study? Please clarify more. 

Response 2: Thank you for the remark. The European Social Survey (ESS) was conducted in Germany and a multitude of other countries throughout Europe and inquired several societal issues, including participants attitude towards climate change. In our study, we used the climate specific items of the ESS to confirm the high climate change awareness German medical students showed in prior studies. We added a remark to clarify that only the climate specific items of the ESS were used in our survey:

„In the eighth round of the European Social Survey (ESS), conducted in 2016, people across Europe were surveyed on social issues, including their thoughts and worries about climate change[35]. The ten questions of the ESS accounting for attitudes towards climate change were identically used in this study to map the medical students’ awareness of climate change and its implications.“ (lines 112-116)

Point 3: Line 127 - did you mean to say item or items?

Response 3: Thank you for the correction. Indeed, it meant to say “items” (now line 135) and we have corrected the respective sentence accordingly.

Point 4: Did you administer these assessments back to back at one setting? Line 126-128. My concern is issues with test-retest reliability and fatigue of exams, then assessments of the subjects. That could be a significant limitation of this study. 

Response 4: Thank you for the remark. Items were asked twice in succession, but the second time they asked specifically about the burden associated with climate change. Therefore, the adapted items differ considerably and don’t represent a retest. Accordingly, test-retest reliability cannot be calculated.  

We added a remark in the section “Strength and Limitations” about the possible effect of fatigue after completion of the exams:

“Fatigue following the completion of their exam could also have an influence on the participants’ results.” (lines 444-445)

Point 5: Line 184 - please further describe and clarify the reason you replaced missing values by the mean?

Response 5: Thank you for the question. We added the following explanation to underpin our rationale:

“For the psychometric measures and the personality factors, 264 (1%) values were missing. Those were replaced by the mean of the item calculated from the available data points, as this method does not overestimate the data’s variance and results are comparable to those obtained when employing multiple imputation methods (Parent, 2013).” (lines 195-198).

Point 6: Lines 273-281 may be better placed in the discussion section instead of the results. You are using result data to then offer other ideas and future research. 

Response 6: Thank you for the observation. We put the mentioned paragraph in a fitting position in the “Discussion” section (now: lines 385-392)

Point 7: Line 319 - you state "over the last years." I would put in the number of years. One item that you may need to also discuss is that the COVID pandemic may have played a role in some of this change in views of climate change and overall distress in society. That may also be a limitation in the data. 

Response 7: Thank you for the remark. We specified “over the last years” to “over the last five years” (line 324). We added a paragraph about the potential effects of the COVID pandemic on our study’s results:

„Another potential distortion of the results of this study is the SARS-CoV-2 pandemic taking place during the survey period, as it has a negative effect on mental health and eclipsed other societal issues such as the ongoing climate crisis at the time of the survey.“ (lines 462-465)

Point 8: Line 417. The assessments you used in this study have been validated. But did you thoroughly validate your modified versions of those assessments?

Response 8: Thank you for the observation. We clarified that the questionnaires that we based our study on are indeed thoroughly validated, yet our use is explorative:

“The presented study is explorative in character, as it employs questionnaires that are validated and widely used in clinical settings to measure depression, anxiety, traumatic symptom burden and perceived stress in an adapted version to measure the extent of the respective symptoms when thinking about climate change.” (lines 445-458).

Round 2

Reviewer 1 Report

The amendments included in the revised document are sufficient and the clarifications provided satisfactory.  

Author Response

Dear Reviewer,

thank you for your valuable remarks and questions regarding our manuscript, helping us considerably to improve our manuscript and clarify the purpose of our research. We are glad we could improve our manuscript sufficiently. 

Sincerely,

Lukas Schwaab, Nadja Gebhardt, Hans-Christoph Friederich and Christoph Nikendei